# Effects of 120 g/h of Carbohydrates Intake during a Mountain Marathon on Exercise-Induced Muscle Damage in Elite Runners

**DOI:** 10.3390/nu12051367

**Published:** 2020-05-11

**Authors:** Aitor Viribay, Soledad Arribalzaga, Juan Mielgo-Ayuso, Arkaitz Castañeda-Babarro, Jesús Seco-Calvo, Aritz Urdampilleta

**Affiliations:** 1Glut4Science, Physiology, Nutrition and Sport, 01004 Vitoria-Gasteiz, Spain; aitor@glut4science.com; 2Institute of Biomedicine (IBIOMED), Physiotherapy Department, University of Leon, Campus de Vegazana, 24071 Leon, Spain; marisolarribal@gmail.com; 3Department of Biochemistry Molecular Biology and Physiology, Faculty of Health Sciences, University of Valladolid, 42004 Soria, Spain; juanfrancisco.mielgo@uva.es; 4Health, Physical Activity and Sports Science Laboratory, Department of Physical Activity and Sports, Faculty of Psychology and Education, University of Deusto, 48007 Bizkaia, Spain; arkaitz.castaneda@deusto.es; 5Institute of Biomedicine (IBIOMED), Physiotherapy Department, University of Leon, Researcher at the Basque Country University, Campus de Vegazana, 24071 Leon, Spain; dr.seco.jesus@gmail.com; 6Centro de Investigacion y de Formación ElikaEsport, 08290 Cerdanyola del Valles, Barcelona, Spain

**Keywords:** dietary intake, muscle recovery, athletic performance, glycogen

## Abstract

Background—exercise-induced muscle damage (EIMD) and internal exercise load are increased after competing in ultraendurance events such as mountain marathons. Adequate carbohydrate (CHO) intake during exercise optimizes athletic performance and could limit EIMD, reduce internal exercise load and, thus, improve recovery. Therefore, the aim of this study was to research into and compare the effects of high CHO intake (120 g/h) in terms of CHO intake recommendation (90 g/h) and regular CHO intake performed by ultraendurance athletes (60 g/h) during a mountain marathon, on exercise load and EIMD markers (creatine kinase (CK), lactate dehydrogenase (LDH), glutamic oxaloacetic transaminase (GOT), urea and creatinine). Materials and Methods—a randomized trial was carried out on 20 male elite runners who had previously undertaken nutritional and gut training, and who consumed different CHO dosages according to experimental (EXP—120 g/h), control (CON—90 g/h) and low CHO intake (LOW—60 g/h) groups during a ~4000 m cumulative slope mountain marathon. EIMD markers were analyzed before the race and 24 h afterwards. Internal exercise load was calculated based on rate of perceived exertion (RPE) during and after the marathon event. Results—internal exercise load during the mountain marathon was significantly lower (*p* = 0.019; η^2^p = 0.471) in EXP (3805 ± 281 AU) compared to LOW (4688 ± 705 AU) and CON (4692 ± 716 AU). Moreover, results revealed that the EXP group evidenced significantly lower CK (*p* = 0.019; η^2^p = 0.373), LDH (*p* < 0.001; η^2^p = 0.615) and GOT (*p* = 0.003; η^2^p = 0.500) values 24 h after the mountain marathon race compared to LOW and CON. Along these lines, EIMD and exercise load evidenced a close correlation (R = 0.742; *p* < 0.001). Conclusion: High CHO intake (120 g/h) during a mountain marathon could limit the EIMD observed by CK, LDH and GOT and internal exercise load compared to CHO ingestion of 60 and 90 g/h.

## 1. Introduction

Ultraendurance events such as mountain marathons (42,195 km) represent one of the major physical challenges for athletes, as they potentially involve various physiological and pathophysiological responses which increase internal exercise load and fatigue [1,2,3]. Considering the variability of the terrain, characteristics and lack of similarity in terms of physical demands when referring to different mountain events, distance by itself does not seem to be a reliable parameter for quantifying physiological and metabolic demands. Other parameters such as physiological intensity, internal exercise load and biochemical changes might represent a better understanding of these event requirements [4,5]. In this sense, internal exercise load has been shown to increase after hard endurance training and competition, representing an augmented internal load and decreased performance [4,6]. This phenomenon could be affected by several psychophysiological and metabolic factors such as rate of perceived exertion (RPE), glycogen depletion, heart rate, dehydration, and exercise-induced muscle damage (EIMD) [3,5,7,8,9,10]. Therefore, monitoring of internal load may reveal the state of fatigue of an athlete and determine a method for systematically quantifying the exercise dose (i.e., work completed), as well as the individual response to training stimulus [11,12] that might act as a guide in the search for different strategies aimed at decreasing it [11].

High-intensity efforts, and particularly eccentric contractions, induce EIMD [13,14,15,16]. This EIMD leads to the onset of an inflammatory response that is associated with, among other factors, deterioration of muscle function, delayed onset muscle soreness (DOMS) and increased muscle metabolism proteins in the blood stream [5,17,18]. Although EIMD markers show an important rise in blood immediately after exercise as a direct consequence of EIMD, higher levels usually appear after 24–72 h, with a total required time of several days to return to baseline values [18,19]. Thus, significant increases in creatine kinase (CK) blood levels have been well documented in athletes following endurance and eccentric exercises such as marathons and ultramarathons [15,19,20,21]. In the same way, an increase in lactate dehydrogenase (LDH) of 37% and 87% after an uphill-only marathon has been documented [21], and after 67 km with approximately 4500 m of a total ascent mountain ultramarathon [1], respectively. In addition, glutamic oxaloacetic transaminase (GOT), blood urea and creatinine have been associated with EIMD and taken on increasing significance following completion of endurance events, increasing as the running distance increased [1,22,23]. 

On the other hand, there is evidence to support the fact that localized glycogen content is associated with muscle function and force production during repeated contractions [24,25]. In this sense, the link between intramyofibrillar glycogen content, the release of Ca^2+^ from the sarcoplasmic reticulum (SR) and the excitation-contraction (E-C) coupling has been shown in elite athletes [26]. Muscle fatigue and EIMD are multifactorial phenomena which involve several factors that are not well established [27]. Among them, it is proposed that metabolic perturbations (e.g., adenosine triphosphate (ADP), Pi, H^+^, reactive oxygen species ROS), impairment of Ca^2+^ release and uptake from the SR and changes in E-C coupling within the muscle participate in these processes [27,28]. As higher levels of glycogen ensure adequate muscle function [26] and lower metabolic perturbations, it is reasonable to hypothesize that maintaining adequate muscle glycogen during exercise could limit EIMD and, therefore, improve recovery. In this sense, providing adequate carbohydrate (CHO) during exercise might represent a satisfactory strategy for maintaining blood glucose concentration and spare muscle glycogen content [29,30], whereas an insufficient supply of glucose could result in hypoglycemia, lower glycogen content and muscle fatigue [31]. In addition, intramyofibrillar glycogen has been shown to have a direct link to insulin-mediated glucose uptake following eccentric exercise, thus underscoring the importance of maintaining adequate glycogen levels and CHO availability during exercise in order to improve recovery and glycogen replenishment [24]. 

Although regular CHO intake by athletes during ultraendurance events is approximately 60 g/h [32], current nutritional recommendations for improving athletic performance in these events is 90 g CHO/h, based on the combination of several intestinal apical transporters (SGLT-1 for glucose and GLUT5 for fructose) [33,34] and with previous nutritional and gut training [35,36]. Nevertheless, taking into consideration several physiological reasons such as translocation of the basolateral glucose transporter, with GLUT2 to the apical membrane in the presence of high glucose concentrations being ≥75 mM [37,38], this could lead us to understand that greater than a 90 g/h CHO intake might be possible and potentially beneficial during endurance sport. Thus, although the effects of >90 g/h CHO intake have been recently researched in literature with controversial results [39,40], Pfeiffer et al. showed that athletes who consumed 120 g/h were among the fastest during 2 ultraendurance events, indicating a delay in the onset of fatigue [41].

Bearing in mind that the physiological demands required in mountain marathons give rise to a high exercise load, EIMD and fatigue, with impaired insulin-mediated glycogen resynthesis in the recovery period [42,43], the hypothesis put forward in this study was that a higher CHO intake than that recommended during exercise could limit EIMD and exercise load, thus improving postexercise recovery. Therefore, the aim of this study was to investigate and compare the effects of high CHO intake (120 g/h), recommended CHO intake (90 g/h) and regular CHO intake (60 g/h) during a mountain marathon on exercise load and EIMD markers (CK, LDH, GOT, urea and creatinine).

## 2. Materials and Methods

### 2.1. Experimental Protocol and Participants

The present study was planned as a randomized trial with the purpose of analyzing the effects of 120 g/h of CHO supplementation on exercise load and EIMD markers (CK, LDH, GOT, urea and creatinine). This CHO intake was compared to international recommendations for events of >3 h (90 g/h) [33,34], and regular athletes’ CHO intake during ultraendurance races (60 g/h) [32]. 

Thirty-one elite male athletes (2 world champions) with at least 5 years of ultratrail experience were recruited for this study. Although neither general nor specific guidelines were provided about gut training in this study, all of the participants carried out personalized gut training (training of the intestinal tract to increase tolerance and absorption capacity) as prescribed by their nutritionists (inclusion criteria). During this gut training, athletes needed to have used CHO intakes of up to 90 g/h at least 2 days/week in the 4 weeks prior to the mountain marathon [35,44]. 

After eliminating 5 participants for not meeting the inclusion criteria (5 years experience in ultradistance events, performing gut training and not taking any medical and performance supplements [45] during the 7 days before to the mountain marathon), the remaining 26 athletes were included in the randomization process. 

The runners were instructed that when they had any injury and/or experienced gastrointestinal discomfort which might compromise their performance, they should withdraw from the race so that this would not influence the results. During the mountain marathon, 6 athletes withdrew (3 with injury and 3 with gastrointestinal problems—reflux and/or flatulence). The remaining runners completed the race without gastrointestinal or injury problems. Therefore, the final sample included in this study comprised 20 athletes, including 2 world champions (6 athletes for the LOW, 7 athletes for CON and 7 athletes for EXP) (Figure 1). 

All runners were examined medically before the study in order to confirm they had no injury or disease. No athletes were suffering from any disease, and none of them took medication. Likewise, to avoid possible interference of other nutritional supplements on EIMD markers, a one-week washout period was also included. In addition, none of the participants used any preworkout supplements on the race day (inclusion criteria).

The 26 enrolled runners were randomized into three different groups using a stratified block design, and an independent statistician put together the randomization sequence using SPSS software as follows—(I) low group that consumed 60 g/h of CHO (LOW; *n* = 8; age: 37.8 ± 9.4 years; height: 175.6 ± 10.3 cm and body mass: 71.8 ± 10.3 kg), (II) control group that consumed 90 g/h of CHO (CON; *n* = 9; age: 37.2 ± 5.4 years; height: 172.3 ± 7.0 cm and body mass: 66.6 ± 10.8 kg) and (III) experimental group that consumed 120 g/h of CHO (EXP; *n* =9; age: 38.0 ± 6.8 years; height: 174.2 ± 3.5 cm and body mass: 67.4 ± 11.1 kg). The three groups took CHO during the mountain marathon via the same 30 g maltodextrin (glucose) and fructose gels (ratio 2: 1) in several flavors (artificially sweetened). The gels were made exclusively for this study at the University of Valladolid Physiology Lab (Soria). This glucose: fructose ratio was used to increase exogenous carbohydrate oxidation during exercise [46]. In this way, glucose is absorbed by the sodium-dependent SGLT1 transporter which is characterized by easily becoming saturated (1 g/min) [47,48]. An excessive CHO intake determines a limitation of CHO absorption and subsequent oxidation [35,47]. On the other hand, fructose is absorbed through the GLUT5 transporter, representing an additional pathway for absorbing CHO [35,49].

The CHO intake protocol was programmed every 15, 20 and 30 min, with participants having to consume ¼, ^1^/_3_ or ½ of the total CHO amount per hour according to EXP, CON, LOW, respectively (Figure 2). On the other hand, the runners did not take any other food apart from these gels, and the athletes only drank water ad libitum during the mountain marathon. 

The official mountain marathon race (42.195 km) took place in Oiartzun (Guipuzcoa-Spain) at a temperature of 10 °C, 60% humidity and wind speed of 10 km/h. There were no major changes in weather conditions while the race was taking place, with the race beginning at 9:00 a.m., and consisting of an entrance and an exit to a circuit that had to be completed three times. The total cumulative slope of the test was 3980.80 m (1990.40 m positive and 1990.40 m negative) (Figure 3), while the maximum height reached during the race was 638.20 m and minimum height 3.80 m. Total mountain marathon time was obtained by official chronometers.

During the event, the heat rate (HR) was recorded continuously throughout the entire test using a GPS HR monitor. Likewise, the average HR (HRM) during the event was also recorded, and race intensity was calculated as (HRM/ HRmax) × 100.

All runners signed a statement of informed consent, and were informed about the experimental procedures, associated risks and the benefits that would be obtained as part of the study. This was a premeditated study in accordance with the Declaration of Helsinki (2008), based on the Fortaleza updated version (2013), and approved by the Human Ethics Committee at the Valladolid Health Area, Valladolid, Spain under number PI 19-1345.

### 2.2. Internal Exercise Load

Internal exercise load was calculated using the individualized Session-RPE method [12,50]. The Session-RPE method was determined as being the product of mountain marathon finishing total time and subsequent RPE (mountain marathon duration × RPE). Using this method, the athlete’s perception of the overall difficulty of the mountain marathon was recorded at the same time as the race ended. The session-RPE scale is based on the Borg category-ratio RPE scale, which translates the athlete’s perception of marathon effort into a numerical score. This test is designed to encourage the athlete to respond to a simple question—how was your workout?—with the goal of obtaining an uncomplicated response that reflects the athlete’s global impression of the workout [50]. 

### 2.3. Blood Sample Analysis

All athletes attended the laboratory for blood extraction at two specific points during the research—(1) the day of the race (T1) and (2) postmountain marathon race (T2—24 h after completing the race). Blood extraction and transportation were performed in accordance with WADA guidelines and all blood samples were collected under basal conditions after a 10–12 h overnight fast at T1 and T2. Ten mL of blood was collected from the antecubital vein with the runners seated in a comfortable situation using a Vacutainer tube with gel and clot activator to obtain serum. Additionally, serum markers of EIMD (CK, LDH and GOT) and protein catabolism markers (urea and creatinine) were assessed using an automatic analyzer (AU5400, Beckman Coulter, Brea, CA, USA).

### 2.4. Dietary Assessment

All diets and menus during the study were prepared individually for each runner by the same experienced and certified nutritionist-dietitian in accordance with international recommendations for ultraendurance sports [33,34]. 

Each of the runners received an individualized diet over a 48 h period before T1 in order to optimize their glycogen content with 9 g CHO/BM/day, 1.5 g/kg BM/day of protein and ~0.5 g fat/kg BM. This diet included, among other things, fish, vegetables and olive oil, but did not include butter or fatty meat in order to avoid any interaction between dietary intake and EIMD [51]. Moreover, at T1, athletes took breakfast at the ElikaEsport Health Center 3 h before the trail marathon race. This breakfast was calculated for each runner as 2 g of CHO/kg of body mass (BM) and was made up of rice, corn cereals with oat beverage, cooked fruit and biscuits with jam or sweet quince, cheese or paste. 

During the official trail marathon event, the runners carried previously configured GPS alarms based on their study group to notify them of the corresponding intake time. In addition, the runners drank water ad libitum during the mountain marathon. 

At the end of the race, each runner consumed the equivalent of 1.2 g of CHO/kg with 0.3 g/kg BW of whey isolate protein in the recovery shakes.

Within the next 24 h until T2, a high CHO diet was consumed by each athlete in order to replenish glycogen levels as much as possible. According to earlier studies, ingesting a high CHO intake (9 g CHO/kg BM/day) could end up restoring almost 90–93% of previous muscle glycogen [52,53]. Therefore, each runner consumed 9 g CHO/kg BM over the following 24 h, with 1.5 g/kg BM of protein and ~0.5 g fat/kg BM. Food type and weight were stablished by the same experienced and certified nutritionist-dietitian.

### 2.5. Statistical Data Analyses

Statistical analysis was performed using SPSS Statistics (SPSS: An IBM Company, version 24.0, IBM Corporation, Armonk, NY, USA) software, with the data being expressed as mean ± standard deviation. 

The Shapiro–Wilk normality test was used for the normalization analysis (*n* < 50) and the Levene test was used to check the uniformity of the variables analyzed. Additionally, statistical significance was designated when *p* < 0.05. 

The percentage changes of the variables studied in each study group between T1 and T2 were calculated as Δ (%): ((T2 − T1) / T1) × 100. Δ (%) of EIMD, and catabolism markers, exercise load, body composition, age and race time were compared across CHO intake groups using one-way ANOVA with the CHO intake groups as the fixed factor. A Bonferroni post hoc test was performed for pairwise comparisons among groups. Bivariate correlations between internal exercise load during study and Δ CK (%) were tested using the Pearson rank order correlation test, and regression line and 95% confidence intervals were also calculated.

Likewise, differences between T1 and T2 in each study variable in each CHO intake group were evaluated using a parametric dependent t-test. Additionally, a two-way repeated measures ANOVA test was carried out to examine interaction effects (time X CHO group) among the CHO intake groups (LOW, CON and EXP) for different EIMD markers.

The effect sizes were performed using partial square eta (η^2^p), and were interpreted according to the one indicating that there is no effect if 0 ≤ η^2^p < 0.05; minimum effect if 0.05 ≤ η^2^p < 0.26; moderate effect if 0.26 ≤ η^2^p < 0.64; and a strong effect if η^2^p ≥ 0.64 [54].

## 3. Results

Table 1 shows the values for the EIMD markers in two study periods (T1 and T2) in the three groups. Significant differences (*p* < 0.05) can be seen in the group-by-time for GOT (*p* = 0.027; η^2^p = 0.363), LDH (*p* <0.001; η^2^p = 0.644) and CK (*p* = 0.032; η^2^p = 0.332). Nevertheless, there were no significant differences in the group-by-time for glucose, urea and creatinine (*p* >0.05). Along these lines, significant increases (*p* < 0.05) between study points were observed for urea and CK in the LOW, CON and EXP; however, for GOT and LDH in the LOW and CON, EXP showed a significant lower GOT, LDH and CK value (*p* < 0.05) regarding LOW and CON at T2.

Table 2 displays RPE, race time, race intensity, HRM and HRmax at the end of race in the three groups. The results did not show any significant differences between groups in RPE (*p* = 0.409; η^2^p = 0.100), race time (*p* = 0.871; η^2^p = 0.018), race intensity (*p* = 0.290; η^2^p = 0.162), HRM (*p* = 0.678; η^2^p = 0.045) and HR max (*p* = 0.334; η^2^p = 0.121). 

Figure 4 shows significant differences in exercise load (*p* = 0.019; η^2^p = 0.471). Specifically, EXP (3805 ± 281 AU) showed significantly lower exercise load than LOW (4688 ± 705 AU) and CON (4692 ± 716 AU). 

The results showed significant differences between groups in the percentages change in GOT change (*p* = 0.003; η^2^p = 0.500), LDH change (*p* < 0.001; η^2^p = 0.615) and CK change (*p* = 0.019; η^2^p = 0.373) (Figure 5). EXP showed a significantly lower increase (*p* < 0.05) in GOT (27.2 ± 23.5%), LDH (8.5 ± 8.5%) and CK (155.9 ± 39.5%) than LOW (GOT: 161.6 ± 98.2%; LDH: 46.7 ± 15.5% and CK: 976.2 ± 631.3%) and CON (GOT: 152.1 ± 64.6%; LDH: 36.7 ± 16.6% and CK: 963.4 ± 713.3%).

Figure 6 shows Pearson’s correlation between exercise load and CK percentage change. There was a significant positive correlation between these parameters, indicating that athletes with greater exercise load showed greater CK changes during the mountain marathon (R = 0.742; *p* < 0.001).

## 4. Discussion

This study was designed to compare the effects of different CHO intakes during a mountain marathon on EIMD markers and exercise load. It was hypothesized that a dose of 120 g/h during a mountain marathon could limit EIMD and exercise load and improve postexercise muscle recovery regarding ultraendurance athletes’ CHO intake (60 g/h) and international recommendations for these events (90 g/h). The main findings obtained from this study revealed that higher CHO intake (120g/h—EXP) than that currently recommended (60—LOW and 90g/h—CON) during a mountain marathon significantly limits the increase in EIMD markers such as CK, LDH and GOT after the event (*p* < 0.05), representing a lower EIMD. In addition, a significantly lower exercise load was found in the EXP group compared to LOW and CON (*p* < 0.05), showing that 120 g/h CHO intake could be a determining factor in the internal exercise load response. These results led us to understand that ingesting a high CHO intake during a mountain marathon event might constitute a suitable strategy for limiting EIMD. Moreover, these findings suggest that current recommendations of 90 g/h CHO for exercise lasting more than 2.5 h might not be enough to limit physiological and metabolic responses after a marathon.

Ultraendurance events represent a major challenge from a physiological and metabolic perspective because of the different terrain requirements that involve both concentric and eccentric muscle contractions [4,55]. Moreover, the duration and intensity of these races mean that a major effort is required, and that these requirements led to significantly increased EIMD markers and exercise load following exercise and, thus, to a compromise in terms of recovery [1,3]. Although several previous studies have found positive effects of ingesting CHO plus protein during exercise on EIMD [56,57,58], CHO dosage used in these studies did not fulfil current nutritional recommendations [47]. This could be a potential limiting factor in understanding the positive results obtained by CHO plus protein regarding solely CHO intake. In fact, the addition of protein to CHO has been shown to increase the speed of glycogen replenishment when suboptimal amounts of CHO have been delivered, helping to reduce EIMD symptoms [59]. However, this work was adjusted to current CHO recommendations with the aim of comparing them to higher intakes [47]. 

EIMD represents one of the main factors of internal fatigue induced by exercise and is characterized by several aspects (alone or combined) such as a decrease in muscle strength, swelling, DOMS and systemic efflux of myocellular enzymes and proteins [60]. Although CK constitutes the most common and widely used biochemical indicator of EIMD [61,62], LDH and GOT are also commonly used for this purpose [61]. Even though these biochemical parameters show a major increase in the bloodstream immediately after exercise, maximum levels have been observed between 24–72 h following this [1,18,19]. In this sense, significant bloodstream CK increases of up to 190% have been documented in athletes after competing in endurance events [15,20,21], up to 370% after eccentric exercise [16,63,64,65] and up to 1447% following a 67 km mountain ultramarathon [1]. Moreover, LDH increases up to 56% have been shown 24 h after completing a mountain marathon [1] and 35% in terms of GOT [22]. These biochemical parameters represent the strong impact of endurance events in EIMD and internal fatigue. Along the same lines, the results obtained in this study have shown a significant increase in EIMD indicators after mountain marathon running, although the results showed significant group-by-time differences in CK, LDH and GOT. In addition, the EXP group evidenced a significantly lower increase in the percentage change of CK, LDH and GOT than the CON and the LOW. These results suggest that CHO intake could play a central role in decreasing the biochemical parameter efflux in the bloodstream, limiting EIMD and internal fatigue [66].

In order to establish a relationship between subjective and objective load, Borresen and Lambert [12] introduced a method based on rating of perceived exhaustion (RPE) and exercising duration. In this sense, exercise load has been shown to increase after hard endurance training and competition, representing an augmented internal load and decreased performance [4,6]. This phenomenon could be explained by the increased RPE and be due to several psychophysiological and metabolic factors such as exercise intensity and duration, EIMD, glycogen depletion, dehydration, environmental conditions and external factors [3,5,7,8,9,10]. In this study, a significantly lower exercise load was found in EXP, compared to LOW and CON. In addition, the results obtained indicated a strong correlation (R = 0.742; *p* < 0.001) between exercise load and CK percentage change during the mountain marathon, suggesting that EIMD had a direct effect on internal load and fatigue. This could be explained by a lower psychophysiological load related to greater ingestion of CHO and the potential effects on delaying muscle glycogen and fatigue, also limiting EIMD [24,59,67].

Muscle glycogen is stored in different locations (subsarcolemmal, intermyofibrillar and intramyofibrillar) around the muscle cell and represents not only an energy store, but also a metabolic, cell signaling and muscle function regulator [25,68]. Intramyofibrillar glycogen plays a key role during repeated contractions by counteracting contractile impairments caused by defective sarcoplasmic reticulum Ca^2+^ release and affected excitation-contraction coupling [25]. Moreover, recent studies have demonstrated that blocking glycogenolytic adenosine triphosphate (ATP) activity leads to impaired muscle function, indicating that a minimum glycogen content should be kept in order to maintain adequate muscle contractions [69]. In addition, glycogen synthesis is impaired after EIMD [42,70,71] and has been associated with reductions in GLUT 4 content and translocation [72], as well as reduced glucose uptake [43,73]. Along these lines, the link between intramyofibrillar glycogen content and the insulin-mediated glucose uptake indicate the importance of maintaining adequate glycogen levels and CHO availability during exercise in order to improve postexercise recovery and glycogen replenishment [24]. Moreover, it is well known that the glucose uptake by the muscle during exercise is increased through the GLUT4 transporter, which is stimulated by muscle contraction, constituting an insulin-independent pathway [74,75]. Therefore, high CHO intake during exercise could represent an opportunity to ensure adequate CHO availability in the muscle cell and to maintain glycogen levels by limiting internal fatigue and EIMD, also improving recovery. Along these lines, although glycogen content was not measured in this study, the lower internal fatigue and EIMD shown in the EXP group compared to LOW and CON could be explained by greater CHO availability and glycogen content following completion of the mountain marathon. 

### 4.1. Limitations, Strengths, and Future Lines

Some methodological limitations of this study concerned the quantification of EIMD response and internal fatigue. Among them is the fact that plasma interleukin-6 (IL-6) was not measured, as this represents a reliable biochemical parameter for assessing inflammation induced by exercise and EIMD [76]. Moreover, C-reactive protein is an acute phase protein produced by the liver in response to IL-6 increases during systemic inflammation and is dependent on several factors such as individual response [77]. Along these lines, it has been demonstrated that CHO intake influences IL-6 but not C-reactive protein following a 32 km mountain trail race [66] and, therefore, it might have been interesting to measure these parameters. In addition, athletes’ glycogen content was not quantified by the authors and, thus, the effects of different CHO intakes on postrace and T2 glycogen content were not established. 

On the other hand, this study presented some strengths. It was completed using a real mountain marathon event with elite athletes, which constitutes a realistic effort by the athletes and reliable research by the authors bearing in mind the complexity of these types of mountain event. Moreover, different CHO amounts (60, 90 and 120 g/h) were ingested in real-life conditions, representing an advantage in our understanding of the field-based limitations and possibilities. Along the same lines, this study has shown that greater ingestion of CHO than that currently recommended (up to 120 g/h) might be well-tolerated following suitable gut nutritional training in the course of such a demanding discipline. 

These findings might open up some relevant future research lines by demonstrating that 120 g/h CHO intake during endurance exercise could be the next CHO target quantity to show possible benefits related to delayed glycogen depletion, limited EIMD and improved performance and recovery. Although current guidelines establish the intestinal absorption limit at 90 g/h with multiple intestinal transporters [36,47,78], it has been demonstrated that 120 g/h might be possibly absorbed without gastrointestinal distress, and that future research is needed to understand the physiological and metabolic mechanisms of this absorption. Along the same lines, this study highlights the need to research into the potential effect of “training the gut” strategy in improving CHO intake, transport and utilization during endurance exercise.

### 4.2. Practical Applications 

This study emphasizes the importance of ingesting CHO during endurance events such as mountain marathons and ultraendurance events. Higher CHO intakes could probably limit metabolic fatigue, EIMD and internal load, thus improving recovery. These could be beneficial for endurance athletes, coaches and nutritionists to enable them to recovery form day-to-day training sessions and, therefore, ensure training capacity and health. As we understand it, every athlete who competes in an endurance event should train in the nutritional aspect and work closely with suitable nutritional strategies to limit EIMD.

## 5. Conclusions

High CHO intake (120 g/h) during a mountain marathon could limit the EIMD observed by CK, LDH and GOT and internal exercise load compared to CHO ingestion of 60 and 90 g/h. 

The effects of this higher CHO intake (120 g/h) compared to the recommended (90 g/h) amount might possibly lead to a new and more suitable strategy to limit EIMD in highly physiological and metabolically demanding endurance exercises such as mountain marathons and ultraendurance events. 

## Figures and Tables

**Figure 1 nutrients-12-01367-f001:**
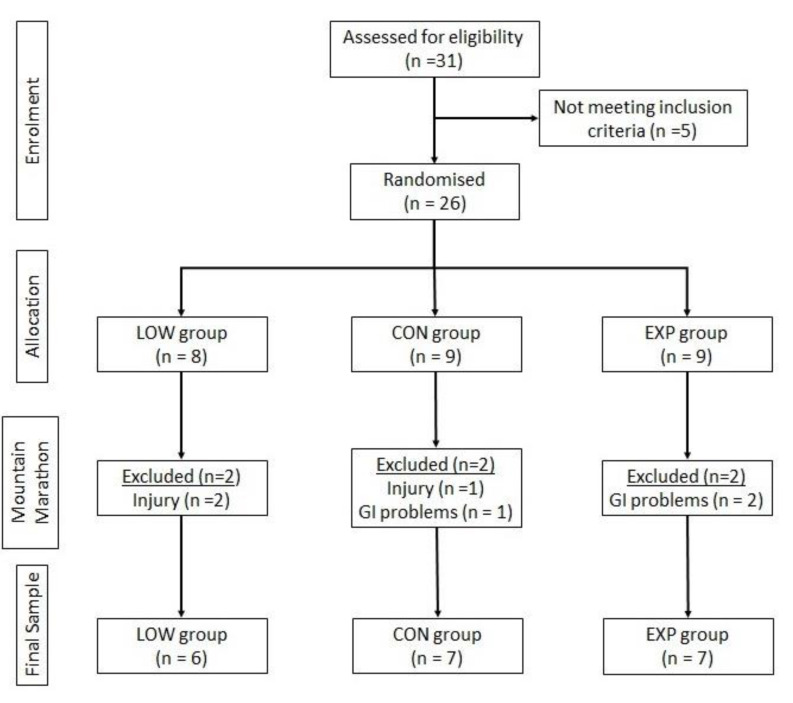
Flow of participants. GI, gastrointestinal.

**Figure 2 nutrients-12-01367-f002:**
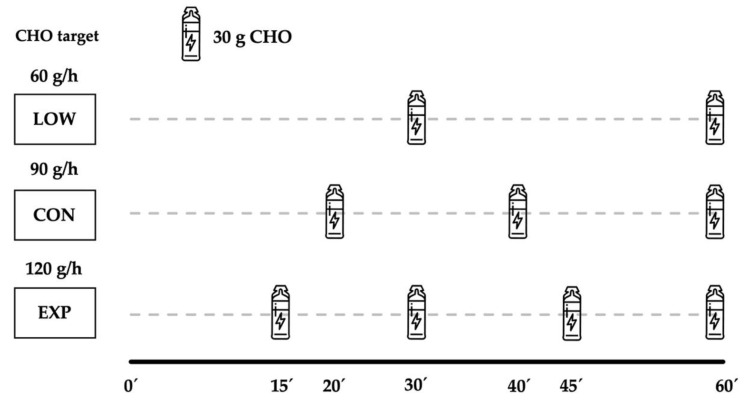
Timing of carbohydrate ingestion during the race for each experimental group.

**Figure 3 nutrients-12-01367-f003:**
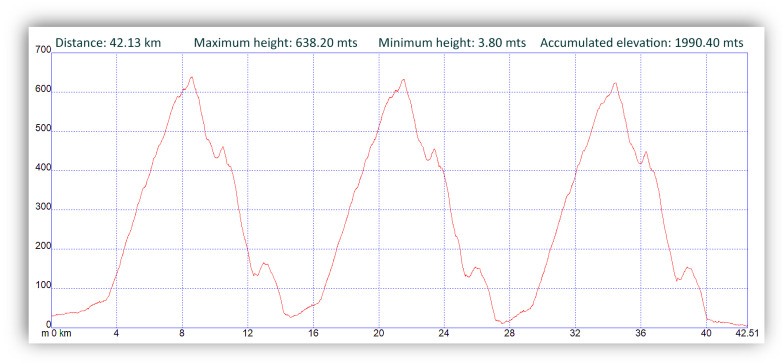
Profile of the trail marathon race.

**Figure 4 nutrients-12-01367-f004:**
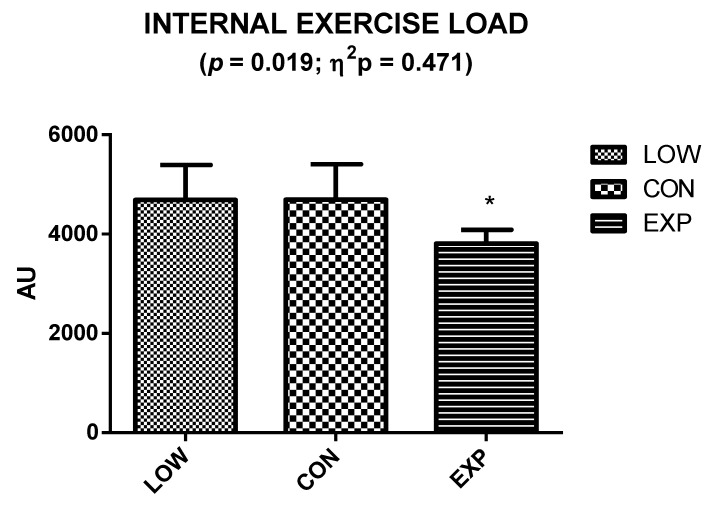
Internal exercise load during the mountain marathon in the different groups. Data are presented as mean ± standard deviation. *p*: Differences by one factor univariant ANOVA tests. * Significant differences from LOW and CON using Bonferroni tests in accordance with one factor univariant ANOVA tests.

**Figure 5 nutrients-12-01367-f005:**
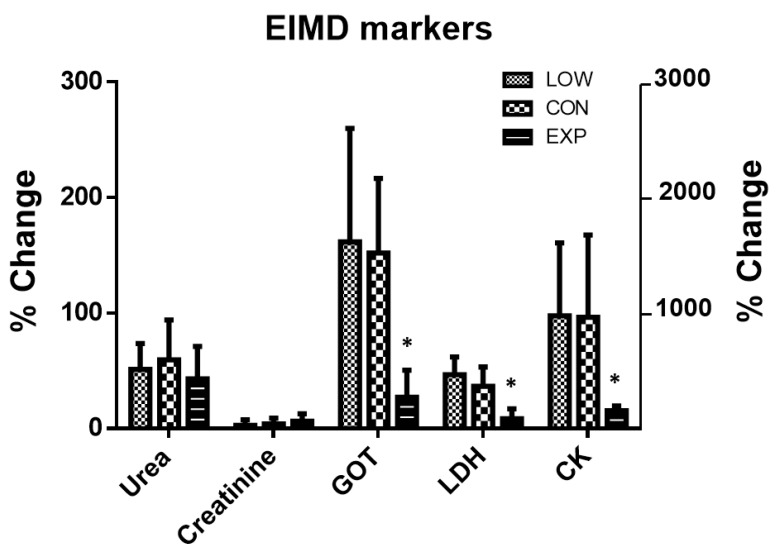
Percentage of EIMD marker changes during the study in the low carbohydrate group (LOW), control group (CON) and experimental group (EXP). Data are presented as mean ± standard deviation. y-axis on the far right indicates % change for creatine kinase (CK) only. * Significant differences from LOW and CON using Bonferroni tests in accordance with one factor univariant ANOVA tests.

**Figure 6 nutrients-12-01367-f006:**
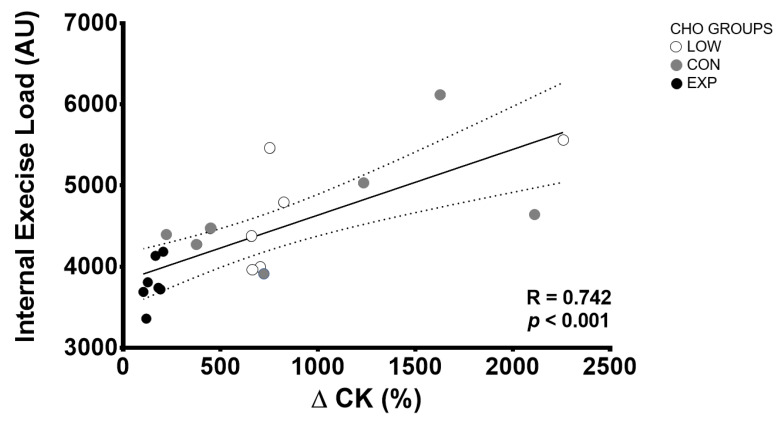
Pearson’s correlation between internal exercise load and CK percentage change.

**Table 1 nutrients-12-01367-t001:** Exercise induced muscle damage (EIMD) markers in the low carbohydrate group (LOW), control group (CON) and experimental group (EXP) before taking part in the event (T1) and after completing it (T2).

	T1	T2	*p*	ŋ^2^p
**Glucose**				
LOW	92.5 ± 8.7	89.7 ± 5.9	0.433	0.099
CON	91.1 ± 5.8	85.6 ± 10.2		
EXP	91.4 ± 7.7	89.3 ± 7.7		
**Urea**				
LOW	28.8 ± 4.0	43.8 ± 10.3 *	0.585	0.061
CON	28.4 ± 4.7	44.6 ± 7.2 *		
EXP	30.9 ± 7.1	42.9 ± 6.1 *		
**Creatinine**				
LOW	0.81 ± 0.08	0.83 ± 0.08	0.606	0.057
CON	0.87 ± 0.08 ^&^	0.91 ± 0.12		
EXP	0.74 ± 0.08	0.79 ± 0.09		
**GOT**				
LOW	26.8 ± 6.0	70.8 ± 36.1 *^&^	0.027	0.363
CON	25.3 ± 6.6	63.0 ± 20.7 *^&^		
EXP	27.5 ± 5.6	36.5 ± 6.5 *		
**LDH**				
LOW	346.2 ± 66.4	504.0 ± 88.9 *^&^	<0.001	0.644
CON	359.1 ± 49.8	489.6 ± 78.3 *^&^		
EXP	381.9 ± 39.7	412.1 ± 29.6		
**CK**				
LOW	137.2 ± 36.0	1528.8 ± 1099.7 *^&^	0.032	0.332
CON	180.4 ± 99.7	1553.0 ± 867.0 *^&^		
EXP	192.9 ± 82.7	499.3 ± 245.8 *		

Data are presented as mean ± standard deviation. *p*: group-by-time interaction (*p* < 0.05). Two-factor repeated-measures ANOVA. * Significant differences (*p* < 0.05) between T1 and T2 in the same CHO group as determined by dependent *t*-tests, ^&^ significant differences (*p* < 0.05) regarding EXP in the same study points using Bonferroni tests.

**Table 2 nutrients-12-01367-t002:** Rate of perceived exertion (RPE), internal exercise load, race intensity, average heart rate (HRM) and maximum heart rate (HRmax) in the low carbohydrate group (LOW), control group (CON) and experimental group (EXP) at baseline (T1).

	LOW	CON	EXP	*p*	ŋ^2^p
RPE	16.2 ± 1.3	16.6 ± 1.6	15.4 ± 1.7	0.409	0.100
Race time	4:38:33 ± 0:43:13	4:44:27 ± 0:40:11	4:31:36 ± 0:41:345	0.871	0.018
Race intensity (%)	82.1 ± 4.6	84.9 ± 2.1	85.4 ± 3.8	0.290	0.162
HRM	159.0 ± 8.9	155.0 ± 9.5	156.9 ± 5.2	0.678	0.045
HRmax	180 ± 9.3	175.5 ± 7.4	182.4 ± 9.3	0.334	0.121

Data are presented as mean ± standard deviation. *p*: Differences by one factor univariant ANOVA tests, and significant differences regarding EXP using Bonferroni tests.

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
