# Peer review of "Effects of 120 g/h of Carbohydrates Intake during a Mountain Marathon on Exercise-Induced Muscle Damage in Elite Runners"

_nutrients, 2020, doi:10.3390/nu12051367_

Round 1
Reviewer 1 Report
In all, a good paper. The authors have done a good job measuring numerous physiological markers in a 'real world' environment. I feel this gives the paper strength over other laboratory based studies.
Comments
English does need tidying throughout. Apart from that, intro was fine.
Methods
Line 112 - I am not sure what gut training is. Potentially consider elaborating if the editor feels appropriate.
Line 129 - remove space after : so readers know its a ratio
Lines 135 and 136 - the timing ("CHO intake protocol was programmed every 15, 20, and 30 minutes...") and the amount ("1/4, 1/3, of 1/2") needs to be clarified. To me, this read that those in the LOW consumed 1/4 CHO every 15 minutes, whilst those in the high consumed 1/2 every 30 mins, but this would result in the same CHO intake. An additional figure about the timings and quantities would help.
Possibly as a supplementary, could group differences for relevant aspects (age, max HR, any basal physiological info) would be good.
The statistical analysis needs to be expanded. There was no mention of correlation etc.
Results and discussion.
Overall, good, and clearly presented.
With the discussion, the strengths, weaknesses, and future direction should be stated as such. I was guessing the first paragraph were the weaknesses, second was strengths, etc - just state up front as such.
Author Response
Point-by-Point Response to Reviewer’s Comments
We would like to sincerely thank the reviewers for their helpful recommendations. We have seriously considered all the comments and carefully revised the manuscript accordingly. Revisions are highlighted in yellow through the manuscript to indicate where changes have taken place. We feel that the quality of the manuscript has been significantly improved with these modifications and improvements based on the reviewers’ suggestions and comments. We hope our revision will lead to an acceptance of our manuscript for publication in Nutrients.
In advance,
King regards
Reviewer 1:
REVIEWER: English does need tidying throughout. Apart from this, intro was fine.
AUTHORS: Thank you for your suggestion. The text has been reviewed by a native English.
REVIEWER: Line 112 – I am not sure what gut training is. Potentially consider elaborating if the editor feels appropriate.
AUTHORS: Thank you for your interest. The authors have included this sentence to clarify the “gut training” concept: “…carried out personalized gut training (training of the intestinal tract to increase tolerance and absorption capacity) as prescribed by their nutritionists (inclusion criteria)”.
REVIEWER: Line 129 – remove space after: so readers know it’s a ratio.
AUTHORS: Thank you for your appreciation. The authors have modified this.
REVIEWER: Line 135 and 136 – the timing (“CHO intake protocol was programmed every 15, 20, and 30 minutes…”) and the amount (“1/4, 1/3, of ½”) needs to be clarified. To me, this read that those in the LOW consumed ¼ CHO every 15 minutes, whilst those in the high consumed ½ every min, but this would result in the same CHO intake. An additional figure about the timings and quantities would help.
AUTHORS: Thank you for your suggestion. The authors have modified it and have included a figure to better understand the timing of the ingestion.
REVIEWER: Possibly as a supplementary, could group differences for relevant aspects (age, max HR, any basal physiological info) would be good.
AUTHORS: Thank you for your suggestion. The authors have added the participants’ age in the Experimental protocol and participants section and heart rate mean and maximum in table 2.
REVIEWER: The statistical analysis needs to be expanded. There was no mention of correlation etc.
AUTHORS: Thank you for your recommendation. The authors included Statistical Data Analyses section information about correlation: “Bivariate correlations between internal exercise load during study and Δ CK (%) were tested using the Pearson rank order correlation test, and regression line and 95% prediction interval for individual runners were also calculated.”
REVIEWER: Strenghts, weaknesses and future directions: state up front as such.
AUTHORS: Thank you for your recommendation. The authors have included in each paragraph of Limitations, strengths, and future lines section clues to let the reader know what we are talking about.

Reviewer 2 Report
The authors have measured the level of muscle damage in elite athletes ingesting variable levels of carbohydrate during a mountain marathon race. The evidence indicates carbohydrate ingestion, beyond that currently recommended, improves indices of muscle damage 24 h following a race. While the dietary control of this study was excellent, the authors have failed to determine the functional outcome of this result and have drawn conclusions that limit its practicality. The authors are directed to more detailed comments below to help improve the clarity of the study.
Major comments
- The argument set forth by the authors in the introduction is not compelling. There are several inconsistencies in references and statements. The literature regarding EIMD is lacking and incorrect at times. The authors are encouraged to re-write this section with an emphasis on the final two paragraphs and to re-consider their use of references throughout. More details are provided below.
- Quantification of exercise load is misleading to the reader, especially given it is calculated from several variables that were not different between groups. Given this is one of the primary outcomes, the authors need to re-consider and re-examine how these values are calculated and disseminated. They also need to explain the adjustments they made with sex because there is not detail in the manuscript about male and female participants.
- The author’s discuss and draw conclusions about how CHO ingestion impacts performance and long-term recovery. The authors did not measure long-term recovery, nor was performance, as indicated by time to finish the race, impacted by CHO ingestion. The authors should re-consider the discussion with specific reference to the outcomes of this study.
- The ultimate measure of performance in this study should either be time to complete a race or muscle performance. In the first instance, performance was not affected by CHO ingestion. In the second instance, the authors failed to measure any functional outcome before and after the race. Therefore, while the study clearly indicates muscle damage after 24 hours was lower with 120 g/h, how this translates to muscular function is unknown.
- There is no detail in the manuscript to disclose how 120g/h affects the participant’s gastrointestinal tract and the suitability of this for athletes in the real-world. Is it safe considering it is well above that currently recommended?
Introduction
- Page 50 – I am not sure what the authors are referring to here. I presume exercise load is referring to “internal exercise load” from the previous sentence? Then in line 52, the authors refer to changes in exercise intensity and biochemical changes but have suggested in line 50 that these changes represent their own parameters with which to measure exercise demands. This section is a little confusing and it may be more appropriate to specifically state what “exercise load” is.
- Line 60 – what are the authors referring to here with “reduced muscle contraction”? Strength? Force production? Fatigability?
- Line 61 – Are the authors calling protein catabolism muscle damage in this instance?
- Sentence structure throughout the manuscript requires editing. The authors should consider recruiting a native English speaker to read the manuscript thoroughly.
- Line 61 – 63 – the authors are stating that urea and creatinine are markers of muscle damage. However, reference 17 does not refer to either in the entire manuscript and reference 19 refers specifically to endurance exercise and never mentions muscle damage or injury. Creatinine is also a simple measure of homeostasis that rises in response to many forms of exercise and may not be associated with EIMD. Therefore, the authors should re-examine the use of specific references in the manuscript and re-consider statements about metabolites indicating EIMD.
- Line 63 – 66 – what markers are the authors referring to here? Creatinine for example, while not a measure of EIMD, increases during and after exercise, not 24-72 hours later.
- Line 74 – Reference 25 states that muscle damage might influence recovery of muscle glycogen, not that glycogen content affects muscle damage.
- Line 78 – 83 – CHO may impact rate of recovery from EIMD, but does it really affect EIMD from occurring as the author’s state?
Methods
- Line 113 – can the authors please define what “supplements” means here?
- Figure 1 – suggest changing the enrolment section to only include n = 5 only once otherwise it indicates exclusion of 10. Also, did GI problems mean they could not finish the race?
- Were subjects able to consume anything else during the race? Was water consumption measured? Could they eat anything other than the CHO supplement provided to them?
- Did the subjects complete any questionnaire about suitability of the CHO supplement for this event?
- Line 142 – 146 – what was the purpose of completing the max aerobic power test? For subject randomisation perhaps? The authors should have considered measuring muscle strength before and after the race as an indicator of EIMD and as a functional measure.
- Line 148 – the conditions did not change during the whole race?
- Line 177 – what was the plasma used for?
- Dietary control was excellent!
- Ensure English and US spelling consistencies. For example, randomisation and randomization are used interchangeably.
Results
- So was their an effect of CHO on muscle damage markers? The results as written do not really describe this.
- Figure 3 – Legend is not required in the graph.
- I do not understand how exercise load can be lower in the EXP group when it is calculated from race time, race intensity and RPE, all of which were not different between groups in Table 2.
- Table 1 and figure 4 show exactly the same results. Why not just show Figure 4? This relates to my first comment above where I was expecting this result first.
- Figure 4 – add another Y-axis for CK or simply adjust the left y-axis to have two sections rather than dividing data by 100.
- How exactly is the data adjusted for sex? How many males and female were there?
- Figure 5 is great, but it indicates the same concern I have for the measurement of exercise load earlier.
Discussion
- The outcomes are very simple and direct. However, the authors measured EIMD markers after 24 h (normally peak 24-72 h post) and have not adequately described exercise load throughout the manuscript. Given this is the primary outcome, more detail is required before drawing conclusions.
- Line 263 – the authors should soften their statements about recovery because this was not measured.
- Line 269 – 270 – the authors did not measure long-term recovery.
- Line 314 – reference 27 refers to muscle fatigue, not EIMD. In this study, muscle glycogen availability did not affect muscle damage because it was never measured.
- Line 359 – “improving performance in the last part of the race and long-term recovery”. The authors did not measure long-term recovery and race time was not different between groups.
- Line 362 – “However, high CHO ingestion levels up to 90-120 g/h could not be possible without previous nutritional and gut training that provide adequate food and liquid tolerance, together with product and CHO type selection.” Because all subjects had “gut training”, the authors cannot state this.
- Line 373 – performance was not improved because of CHO.
Author Response
Point-by-Point Response to Reviewer’s Comments
We would like to sincerely thank the reviewers for their helpful recommendations. We have seriously considered all the comments and carefully revised the manuscript accordingly. Revisions are highlighted in yellow through the manuscript to indicate where changes have taken place. We feel that the quality of the manuscript has been significantly improved with these modifications and improvements based on the reviewers’ suggestions and comments. We hope our revision will lead to an acceptance of our manuscript for publication in Nutrients.
In advance,
King regards
Reviewer 1:
The authors have measured the level of muscle damage in elite athletes ingesting variable levels of carbohydrate during a mountain marathon race. The evidence indicates carbohydrate ingestion, beyond that currently recommended, improves indices of muscle damage 24 h following a race. While the dietary control of this study was excellent, the authors have failed to determine the functional outcome of this result and have drawn conclusions that limit its practicality. The authors are directed to more detailed comments below to help improve the clarity of the study.
Major comments
REVIEWER: The argument set forth by the authors in the introduction is not compelling. There are several inconsistencies in references and statements. The literature regarding EIMD is lacking and incorrect at times. The authors are encouraged to re-write this section with an emphasis on the final two paragraphs and to re-consider their use of references throughout. More details are provided below.
AUTHORS: Thank you for your recommendation. The authors have re-written this section and emphasized on the final two paragraphs following the specific reviewer’s recommendations. Likewise, the authors have modified some references throughout.
REVIEWER: Quantification of exercise load is misleading to the reader, especially given it is calculated from several variables that were not different between groups. Given this is one of the primary outcomes, the authors need to re-consider and re-examine how these values are calculated and disseminated. They also need to explain the adjustments they made with sex because there is not detail in the manuscript about male and female participants.
AUTHORS: Thank you for your indication. The authors have explained adequately how internal exercise load was calculated in Internal exercise load section.: “Internal exercise load was calculated using the individualized Session-RPE method [12,50]. The Session-RPE method was determined as being the product of mountain marathon finishing total time and subsequent RPE (mountain marathon duration x RPE).”
On the other hand, the authors have deleted from table 1 footer “EIMD results were adjusted according to sex” because all the participants were males. Moreover, the participants’ sex has been included in the experimental protocol and participants section: “Thirty-one elite male athletes (2 world champions) with at least 5 years of ultra-trail experience were recruited for this study.”
REVIEWER: The author’s discuss and draw conclusions about how CHO ingestion impacts performance and long-term recovery. The authors did not measure long-term recovery, nor was performance, as indicated by time to finish the race, impacted by CHO ingestion. The authors should re-consider the discussion with specific reference to the outcomes of this study.
AUTHORS: Thank you for your indication. The authors have modified the conclusion following the reviewer’s indication:
“High CHO intake (120 g/h) during a mountain marathon could limit the EIMD observed by CK, LDH and GOT and internal exercise load compared to CHO ingestion of 60 and 90 g/h.
“The effects of this higher CHO intake (120 g/h) compared to the recommended (90 g/h) amount might possibly be lead to a new and more suitable strategy to limit EIMD in highly physiological and metabolically demanding endurance exercises such as mountain marathons and ultra-endurance events. “
REVIEWER: The ultimate measure of performance in this study should either be time to complete a race or muscle performance. In the first instance, performance was not affected by CHO ingestion. In the second instance, the authors failed to measure any functional outcome before and after the race. Therefore, while the study clearly indicates muscle damage after 24 hours was lower with 120 g/h, how this translates to muscular function is unknown.
AUTHORS: Thank you for your comment. Indeed, the authors have not shown benefits in performance and muscle function with the study parameters examined. Therefore, the authors have eliminated any indication in this regard throughout the text.
REVIEWER: There is no detail in the manuscript to disclose how 120g/h affects the participant’s gastrointestinal tract and the suitability of this for athletes in the real-world. Is it safe considering it is well above that currently recommended?
AUTHORS: Thank you for your interest. One of the inclusion criteria was that the runners should had performed a gut training. In this regard, the authors have indicated in Experimental protocol and participants section: “. Although neither general nor specific guidelines were provided about gut training in this study, all of the participants carried out personalized gut training (training of the intestinal tract to increase tolerance and absorption capacity) as prescribed by their nutritionists (inclusion criteria). During this gut training, athletes needed to have used CHO intakes of up to 90 g/h at least 2 days/week in the 4 weeks prior to the mountain marathon [35,44].”
Moreover, the authors have indicated in the same section that “. During the mountain marathon, 6 athletes withdrew (3 with injury and 3 with gastrointestinal problems – reflux and/or flatulence-). The remaining runners completed the race without gastrointestinal or injury problems.”
Lastly, the authors included in Limitations, strengths, and future lines section: “Along the same lines, this study has shown that greater ingestion of CHO than that currently recommended - up to 120 g/h - might be well-tolerated following suitable gut nutritional training in the course of such a demanding discipline.”
Introduction:
REVIEWER: Page 50 – I am not sure what the authors are referring to here. I presume exercise load is referring to “internal exercise load” from the previous sentence? Then in line 52, the authors refer to changes in exercise intensity and biochemical changes but have suggested in line 50 that these changes represent their own parameters with which to measure exercise demands. This section is a little confusing and it may be more appropriate to specifically state what “exercise load” is.
AUTHORS: Thank for your indication. The authors referred to “internal exercise load” and, therefore, it has been modified in order to better understand this concept. The new sentence is: “This phenomenon could be affected by several psychophysiological and metabolic factors such as rate of perceived exertion (RPE), glycogen depletion, heart rate, dehydration, and exercise induced muscle damage (EIMD) [3,5,7–10].”
REVIEWER: Line 60 – what are the authors referring to here with “reduced muscle contraction”? Strength? Force production? Fatigability?
AUTHORS: Thank you for your appreciation. The authors have modified this sentence adding the word “capacity” to muscle contraction.
REVIEWER: Line 61 – Are the authors calling protein catabolism muscle damage in this instance?
AUTHORS: Thank you for your observation. The authors have deleted this sentence in order to avoid confusion.
REVIEWER: Sentence structure throughout the manuscript requires editing. The authors should consider recruiting a native English speaker to read the manuscript thoroughly.
AUTHORS: Thank you for your suggestion. The text has been reviewed by a native English.
REVIEWER: Line 61 – 63 – the authors are stating that urea and creatinine are markers of muscle damage. However, reference 17 does not refer to either in the entire manuscript and reference 19 refers specifically to endurance exercise and never mentions muscle damage or injury. Creatinine is also a simple measure of homeostasis that rises in response to many forms of exercise and may not be associated with EIMD. Therefore, the authors should re-examine the use of specific references in the manuscript and re-consider statements about metabolites indicating EIMD.
AUTHORS: Thank you for your appreciation. The authors agree with your comments and have deleted this sentence in order to avoid confusion.
REVIEWER: Line 63 – 66 – what markers are the authors referring to here? Creatinine for example, while not a measure of EIMD, increases during and after exercise, not 24-72 hours later.
AUTHORS: Thank you for your comment. After eliminating the previous sentence, the authors hope this is clarified.
REVIEWER: Line 74 – Reference 25 states that muscle damage might influence recovery of muscle glycogen, not that glycogen content affects muscle damage.
AUTHORS: Thank you for your observation. The authors agree with you and have changed the sentence with more precise information.
On the other hand, there is evidence to support that localized glycogen content is associated with muscle function and force production [24,25].
REVIEWER: Line 78 – 83 – CHO may impact rate of recovery from EIMD, but does it really affect EIMD from occurring as the author’s state?
AUTHORS: Thank you for your appreciation. The authors have included more information in the manuscript to justify this relation.
“On the other hand, there is evidence to support the fact that localized glycogen content is associated with muscle function and force production [24,25]. In this sense, the link between intramyofibrillar glycogen content, the release of Ca2+ from the sarcoplasmic reticulum (SR) and the excitation-contraction (E-C) coupling has been shown in elite athletes [26]. Muscle fatigue and EIMD are multifactorial phenomena which involve several factors that are not well established [27]. Among them, it is proposed that metabolic perturbations (e.g. ADP, Pi, H+, ROS), impairment of the Ca2+ release and uptake from the SR and changes in E-C coupling within the muscle participate in these processes [27,28]. As higher levels of glycogen ensure adequate muscle function [26] and lower metabolic perturbations, it is reasonable to hypothesize that maintaining adequate muscle glycogen during exercise could limit EIMD and, therefore, improve recovery. In this sense, providing adequate CHO during exercise might represent a satisfactory strategy for maintaining blood glucose concentration and spare muscle glycogen content [29,30], whereas an insufficient supply of glucose could result in hypoglycemia, lower glycogen content and muscle fatigue [31]. In addition, intramyofibrillar glycogen has been shown to have a direct link to insulin-mediated glucose uptake following eccentric exercise, thus underscoring the importance of maintaining adequate glycogen levels and CHO availability during exercise in order to improve recovery and glycogen replenishment [24]. “
Methods:
REVIEWER: Line 113 – can the authors please define what “supplements” means here?
AUTHORS: Thank you for your observation. The authors have changed the terminology according to AIS: “After eliminating 5 participants for not meeting the inclusion criteria ( 25 years’ experience in ultra-distance events, performing gut training and not taking any medical and performance supplements [45] during the 7-days before to the mountain marathon), the remaining 26 athletes were included in the randomization process.”
REVIEWER: Figure 1 – suggest changing the enrolment section to only include n = 5 only once otherwise it indicates exclusion of 10. Also, did GI problems mean they could not finish the race?
AUTHORS: Thank you for your suggestion. The figure has been modified. Regarding your second question authors included: “During the mountain marathon, 6 athletes withdrew (3 with injury and 3 with gastrointestinal problems – reflux and/or flatulence-).”
The runners were instructed that when they had any injury and/or gastrointestinal discomfort that compromise their performance, they should withdraw from the race so that this fact did not influence the results.
REVIEWER: Were subjects able to consume anything else during the race? Was water consumption measured? Could they eat anything other than the CHO supplement provided to them?
AUTHORS: Thank for your indication. The authors have included the following sentence: “On the other hand, the runners did not take any other food than these gels. The athletes only drank water ad libitum during the mountain marathon”.
REVIEWER: Did the subjects complete any questionnaire about suitability of the CHO supplement for this event?
AUTHORS: Thank for your observation. No questionnaire was completed but every athlete who finished the race declared no gastrointestinal and injury problems. The authors have included this sentence: The remaining completed the race with no gastrointestinal and injury problems.
REVIEWER: Line 142 – 146 – what was the purpose of completing the max aerobic power test? For subject randomisation perhaps? The authors should have considered measuring muscle strength before and after the race as an indicator of EIMD and as a functional measure.
AUTHORS: Thank for your comment. Because of the max aerobic power test does not have relevance in this manuscript, the authors have removed it. On the other hand, thank you for suggesting the measure of the strength pre and post. Regarding this, we have now under review another paper with these data.
REVIEWER: Line 148 – the conditions did not change during the whole race?
AUTHORS: Thank for your observation. Very little variation of conditions was registered during the race. Anyway, the authors have included a sentence explaining this: ”There were no major changes in weather conditions while the race was taking place,….”
REVIEWER: Line 177 – what was the plasma used for?
AUTHORS: The authors thank your appreciation. It has been removed in order to avoid misunderstandings.
REVIEWER: Dietary control was excellent!
AUTHORS: The authors thank your comment.
REVIEWER: Ensure English and US spelling consistencies. For example, randomisation and randomization are used interchangeably.
AUTHORS: The authors thank your appreciation. The text has been reviewed by an English native and these terms have been adjusted as suggested.
Results:
REVIEWER: So was their an effect of CHO on muscle damage markers? The results as written do not really describe this.
AUTHORS: The authors thank your observation. The authors have added new information in the “Results” section regarding the effects of CHO intake in EIMD markers. “Along these lines, although significant increases (p < 0.05) between study points were observed for urea and CK in the LOW, CON and EXP; while for GOT and LDH in the LOW and CON, EXP evidenced a significant lower GOT, LDH and CK value (p < 0.05) regarding LOW and CON at T2.”
REVIEWER: Figure 3 – Legend is not required in the graph.
AUTHORS: Thank for your correction. The authors have removed the legend.
REVIEWER: I do not understand how exercise load can be lower in the EXP group when it is calculated from race time, race intensity and RPE, all of which were not different between groups in Table 2.
AUTHORS: Thank for your observation. Although no significant differences were observed on marathon time and RPE, tendency was clearly lower in EXP. When we performed internal exercise load by Session-RPE method (mountain marathon duration x RPE), the statistics showed a significant lower internal exercise load in EXP than CON and LOW.
REVIEWER: Table 1 and figure 4 show exactly the same results. Why not just show Figure 4? This relates to my first comment above where I was expecting this result first.
AUTHORS: Thank your appreciation. The authors understand that information in Table 1 is important, because they are absolute values of EIMD vs % of changes values presented in figure 4. Anyway, statistical significance among groups by Bonferroni test has been included in table 1.
REVIEWER: Figure 4 – add another Y-axis for CK or simply adjust the left y-axis to have two sections rather than dividing data by 100.
AUTHORS: Thank your suggestion. The authors have added another Y-axis for CK, as you have proposed.
REVIEWER: How exactly is the data adjusted for sex? How many males and female were there?
AUTHORS: The authors thank your appreciation. The description of the Table 1 according to the sex was an error, when copying it from other authors previous work. All the participants were males and it has been included in the methods section: “Thirty-one elite male athletes (2 world champions) with at least 5 years of ultra-trail experience were recruited for this study.”
REVIEWER: Figure 5 is great, but it indicates the same concern I have for the measurement of exercise load earlier.
AUTHORS: Thank for your observation. The authors have included more information about internal exercise load. Moreover, with the modifications done through the manuscript, the authors hope to be clarified.
Discussion:
REVIEWER: The outcomes are very simple and direct. However, the authors measured EIMD markers after 24 h (normally peak 24-72 h post) and have not adequately described exercise load throughout the manuscript. Given this is the primary outcome, more detail is required before drawing conclusions.
AUTHORS: The authors thank your appreciation. More information about internal exercise load and its calculation has been included in the Internal exercise load section.
REVIEWER: Line 263 – the authors should soften their statements about recovery because this was not measured.
AUTHORS: The authors thank your appreciation. This sentence has been removed.
REVIEWER: Line 269 – 270 – the authors did not measure long-term recovery.
AUTHORS: The authors thank your observation. The authors have removed “long-term recovery”.
REVIEWER: Line 314 – reference 27 refers to muscle fatigue, not EIMD. In this study, muscle glycogen availability did not affect muscle damage because it was never measured.
AUTHORS: The authors thank your appreciation. To avoid incorrect information, EIMD has been removed from this sentence.
REVIEWER: Line 359 – “improving performance in the last part of the race and long-term recovery”. The authors did not measure long-term recovery and race time was not different between groups.
AUTHORS: Thank for your comment. The authors have proposed this practical application based on the hypothesis that CHO ingestion could limit EIDM and, therefore, improve recovery. However, the authors have removed some sentences about performance.
REVIEWER: Line 362 – “However, high CHO ingestion levels up to 90-120 g/h could not be possible without previous nutritional and gut training that provide adequate food and liquid tolerance, together with product and CHO type selection.” Because all subjects had “gut training”, the authors cannot state this.
AUTHORS: The authors thank your appreciation and have removed this state from the manuscript.
REVIEWER: Line 373 – performance was not improved because of CHO.
AUTHORS: The authors thank your appreciation. This sentence has been removed.

Round 2
Reviewer 2 Report
Thank you to the authors for responding to my comments with great clarity. I have a few further comments for them to address based on the original ones.
Line 60 – muscle capacity still does not tell the reader what this is. I suggest the authors read the reference and specifically state what changed.
Line 72 – are the author’s sure glycogen content affects force production? Perhaps over multiple contractions but certainly not for a single tetanic contraction.
Line 113 – is it 5 years’ experience or 25 years (line 119)
Figure 2 caption – suggest changing this to “Timing of carbohydrate ingestion during the race for each experimental group”.
For all figures – can I make a suggestion to make the symbols easier to understand. Figure 4 for example, EXP should have the * above it and the footnote should simply read, “* different from LOW and CON”.
Figure 5 – the data is presented better, but I will let the authors decide whether this is a suitable approach. Perhaps putting the second y-axis on the far right of the graph and indicate % change for CK only.
Figure 6 – this should really be made in graphpad as other figures have been.
Author Response
Point-by-Point Response to Reviewer’s Comments
We would like to sincerely thank the reviewer for their helpful recommendations again. We have seriously considered all the comments and carefully revised the manuscript accordingly. Revisions are highlighted in green through the manuscript to indicate where changes have taken place. We feel that the quality of the manuscript has been significantly improved with these modifications and improvements based on the reviewers’ suggestions and comments. We hope our revision will lead to an acceptance of our manuscript for publication in Nutrients.
In advance,
King regards
Reviewer 2
REVIEWER: Line 60 – muscle capacity still does not tell the reader what this is. I suggest the authors read the reference and specifically state what changed.
AUTHORS: Thank you for your recommendation. The authors have changed muscle capacity by “deterioration of muscle function”.
REVIEWER: Line 72 – are the author’s sure glycogen content affects force production? Perhaps over multiple contractions but certainly not for a single tetanic contraction.
AUTHORS: Thank you for your observation. The authors have changed force production by “during repeated contractions”.
REVIEWER: Line 113 – is it 5 years’ experience or 25 years (line 119)
AUTHORS: Thank you for your observation. The authors have corrected 25 years by 5 years.
REVIEWER: Figure 2 caption – suggest changing this to “Timing of carbohydrate ingestion during the race for each experimental group”.
AUTHORS: Thank you for your suggestion. The authors have included “Timing of carbohydrate ingestion during the race for each experimental group”.in the figure 2 caption.
REVIEWER: For all figures – can I make a suggestion to make the symbols easier to understand. Figure 4 for example, EXP should have the * above it and the footnote should simply read, “* different from LOW and CON”.
AUTHORS: Thank you for your suggestion. The authors have included the “*” above EXP in figure 4 and figure 5. Likewise, the authors have included in the footnote: “*: Significant differences from LOW and CON using Bonferroni tests in accordance with one factor univariant ANOVA tests.”
REVIEWER: Figure 5 – the data is presented better, but I will let the authors decide whether this is a suitable approach. Perhaps putting the second y-axis on the far right of the graph and indicate % change for CK only.
AUTHORS: Thank you for your recommendation. The authors have added the second y-axis on the far right of the graph and have indicated % change for CK only in the footnote.
REVIEWER: Figure 6 – this should really be made in graphpad as other figures have been.
AUTHORS: Thank you for your suggestion. The authors have made figure 6 in Graphpad Prism.
Figure 6. Pearson´s correlation between internal exercise load and CK percentage change. |
